# Assessment of FreeStyle Libre Flash Glucose Monitoring System Implementation in Real Life Clinical Setting: A Prospective Observational Study

**DOI:** 10.3390/diagnostics11020305

**Published:** 2021-02-13

**Authors:** Silvija Canecki Varzic, Kristina Steiner, Maja Gradinjan Centner, Mladen Grgurevic, Maja Cigrovski Berkovic, Sanja Klobucar Majanovic, Maja Radman, Tanja Milicevic, Dario Rahelic, Ines Bilic Curcic

**Affiliations:** 1Department of Endocrinology and Metabolism Disorders, Clinical Hospital Center Osijek, 31000 Osijek, Croatia; silvija.canecki@gmail.com (S.C.V.); steiner.kristina2601@gmail.com (K.S.); majagradinjan@gmail.com (M.G.C.); 2Faculty of Medicine, University J.J.Strossmayer, 31000 Osijek, Croatia; 3Vuk Vrhovac University Clinic for Diabetes, Endocrinology and Metabolic Diseases, Merkur University Hospital, Dugi dol 4a, 10000 Zagreb, Croatia; grgurevic.mladen@gmail.com (M.G.); dario.rahelic@gmail.com (D.R.); 4Department for Endocrinology, Diabetes and Metabolism Clinical Hospital Dubrava, 10000 Zagreb, Croatia; maja.cigrovskiberkovic@gmail.com; 5Department for Endocrinology, Diabetes and Metabolism University Hospital Center Rijeka, 51000 Rijeka, Croatia; sanja.klobucarm@gmail.com; 6Department for Endocrinology, Diabetes and Metabolism University Hospital Center Split, 21000 Split, Croatia; maja.radman1@st.t-com.hr (M.R.); tanja.milicevic2@gmail.com (T.M.)

**Keywords:** FreeStyle Libre, diabetes type 1, real life settings, glycemic control, education

## Abstract

Background: In this study, we investigated the effectiveness of FreeStyle Libre Flash Glucose Monitoring (FGM) implementation in a real life clinical setting with the emphasis on the effect of initial education on the use of the FGM system. Methods: This prospective observational study included 425 diabetes type 1 subjects followed up for 3 to 12 months (320 were followed up to 3 months, 267 up to 6 months and 147 up to period of one year). An FGM sensor was placed at study entry and all participants were educated through a period of 5 days on sensor usage and self-management of glycemia with follow up visits every 3 months. Results: HbA1c values significantly decreased from baseline (T0) to 3 months (T3) (*p* < 0.001), with a drop from 7.48% ± 0.1% to 7.30 ± 0.1%. There was no change in time spent in hypoglycemia from T3 to T12, although there was a decreasing trend present. The change in HbA1c values in the entire cohort was driven by change in the subgroup of patients with HbA1c ≥7% with a drop from 8.22% ± 1.14% to 7.68% ± 1.26% (*p* < 0.0001) in the first 3 months. Also, in individuals performing SMBG less than 5 times per day, there was a steady decrease in HbA1c levels up to 6 months (*p* < 0.05 and *p* < 0.001, respectively) as opposed to those who performed SMBG ≥5 times per day. Conclusions: The improvement in HbA1c was mainly driven by the increase in the number of scans per day. The subjects with poorer glycemic control and those who seldom performed SMBG benefited the most.

## 1. Introduction

Glycemic management aiming at near normal blood glucose concentrations is a key for successful diabetes care but can sometimes impose a risk of hypoglycemia and a need for frequent self-monitoring of blood glucose (SMBG) by finger pricking. HbA1c, a traditionally used marker of glucose control does not give information about intraday or intraday glycemic variability, nor on postprandial hyper- or hypoglycemia, and can therefore be misleading and insufficient in case therapy adjustments are needed. Flash glucose monitoring (FGM) provides on-demand interstitial glucose readings, glucose trend information, estimation of HbA1c and moreover, historical data that can be used to improve diabetes self-management [1]. Glucose readings are stored every 15 min as long as a patient scans the sensor at least once every 8 h. Its ease of use and convenience for patients have made the FGM widely used in type 1 diabetic patients. Growing clinical evidence supports its relevance in terms of hypoglycemia reductions and improvements in time spent in optimal glucose range [1,2,3,4,5]. Data obtained by FGM are generated as AGP (Ambulatory Glucose profile) reports providing information on glucose variability and frequency and duration of hypoglycemic events [6]. AGP reports are easy to use and may serve as an educational tool for patients alone or with health care professionals. 

Regarding a change in HbA1c data are somewhat controversial. In the landmark RCTs of FGM efficacy a change in HbA1c has not been observed [1,2]. On the other hand, the majority of observational studies conducted in routine clinical practice have reported a significant reduction in HbA1c with the usage of FGM [7,8,9,10,11,12,13,14,15]. Furthermore, recently published studies suggested that patients with good glycemic control have no benefits from FGM use, while those with suboptimal control tend to reach significant improvement in HbA1c levels probably due to the higher motivation and empowerment of patients in therapeutic decision making [8,10,11,16].

The simplicity, ease of use, and reliability of FGM provides confidence in users enabling them to reduce the number of finger prick tests, one of the main reasons of discomfort and distress in diabetic patients [17]. Consequently, the number of sensor readings per day increases resulting in improved glycemic control and time in range (TIR) [18]. The mentioned advantages of FGM finally lead to positive impact on psychological wellbeing and improved quality of life [10,11,16,19].

The aim of this study was to assess the usefulness and effectiveness of FreeStyle Libre flash glucose monitoring system implementation in a real life clinical setting in diabetes type 1 patients, with the emphasis on the effect of initial education on the use of the FGM system and the adjustment of therapy based to CGM metrics. 

## 2. Study Design and Patients

This prospective observational study was initiated as the FGM technology became available in Croatia (September 2018.). 425 subjects with type 1 diabetes were recruited from 5 Hospital Clinical Centres (two from Zagreb and one from Osijek, Rijeka and Split) from September 2018 until March 2020 followed up for a minimum of 3 months up to a period of one year (320 were followed up to 3 months, 267 up to 6 months and 147 up to period of one year). The inclusion criteria were type 1 diabetes diagnosed for more than 3 months, subjects aged 18 years or older. All participants provided informed consent and the study was approved by local Ethics Committee (No: R2-12487/2019) and was conducted in accordance with the Declaration of Helsinki. Given the observational nature of the study, no specific exclusion criteria were implemented except pregnancy. Baseline characteristics were recorded from each participant at initiation visit, including duration of diabetes, HbA1c (local laboratory), number of SMBGs per day, type of insulin delivery, and presence of hypoglycemia in the last three months defined as symptomatic with BG < 3.9 mmol/L, severe (requiring assistance of other person or glucagon injection), nocturnal (BG level < 3.9 mmol/L from 10 p.m. to 6 a.m.) and hypoglycemia unawareness (defined as measured BG < 3.9 mmol/L without symptoms). All participants were educated through a period of 5 days on sensor usage and self-management of insulin therapy according to sensor readings, while performing finger prick BG measurement was encouraged in case of hypoglycemia and abrupt changes in glucose levels readings following trend arrows. Scanning of sensor was recommended as many times as deemed necessary to optimize glycemic control and avoid hypoglycemia, no less than 10, optimally at least 15 scans per day. Since the study was observational and we wanted to gain insight into changes in glycemic control as well as self-management of diabetes in the actual clinical practice, follow up visits were scheduled every 3 months (after 3, 6, and 12 months; T3, T6 and T12, respectively), as is common in every day clinical practice, taking into account the problem of unavailability of specialist health care if follow up visits would be performed every 2 weeks. At each visit, glucose data were uploaded from the reader and AGP-derived reports were generated, data were than analyzed together with patient and education was provided on how to achieve more stringent control with minimum of hypoglycemia.

The subjects were then divided in several groups to assess which group of patients benefited the most from FGM placement: first the entire cohort was analyzed, and then divided according to HbA1c (< or ≥7%), number of SMBGs performed prior FGM placement (< or ≥ 5 per day), type of therapy (MDI or insulin pump) and duration of diabetes (< or ≥10 years). 

## 3. Statistical Analysis

Elementary statistics considering means, quartiles, and standard deviations were calculated for continuous variables. Categorical variables were presented with frequency tables and cross tables. 

For testing differences in means when comparing two time periods, the t-test for dependent samples was used. In order to compare means in T0, T3, T6 and T12 or T3, T6 and T12, repeated measures ANOVA is applied. χ^2^ test was used for testing dependence between two categorical variables. Correlation analysis was performed using Pearson test. 

Statistical analyses were performed by using statistical software Statistica 13.5.

## 4. Results 

### 4.1. Baseline Characteristics of Subjects 

Of 425 subjects included in the study, 320 were followed up to 3 months, 267 up to 6 months and 147 up to period of one year. The average age of patients was 41 ± 14.58 years, duration of diabetes 20 ± 12.07 years with average HbA1c of 7.57 ± 1.41%. 95 subjects (20%) were treated with insulin pump therapy, while 354 (76.7%) of them were using multiple daily injections (MDI). Median number of SMBG controls was 5 with a minimum of 2 up to 20 measurements per day. Table 1. 

The majority of patients had symptomatic hypoglycemic events, around half reported nocturnal hypoglycemia, while 58.64% had asymptomatic hypoglycemic events. Table 1. 

### 4.2. HbA1c and Other Parameters of Glucoregulation Assessed by FGM 

In the entire cohort, HbA1c values significantly decreased from baseline (T0) to 3 months (T3) (*p* < 0.001), with a drop from 7.48% ± 0.1% to 7.30 ± 0.1%, while no change was observed afterwards (*p* = NS) from T3 to T12. Time above range (TAR) decreased significantly (T3 40.89% ± 1.79%; T6 37.31% ± 1.71%; T12 34.38% ± 1.62%; *p* < 0.0001), while time in range (TIR) increased significantly (T3 49.94% ± 1.74%; T6 53.75% ± 1.65%; T12 57.07% ± 1.62%; *p* < 0.0001) over time from T3 to T12. However, there was no change in time spent in hypoglycemia (TBR; time below range) in the same follow up period, although there was a decreasing trend present (T3 9.22% ± 0.86%; T6 8.94% ± 0.59%; T12 8.35% ± 0.51%; *p* = NS). In addition, no changes were noted in number and duration of hypoglycemic events (data not shown). Almost all subjects experienced decreased SMBG controls and started scanning more frequently as soon as the FGM was placed. Consequently, the number of scans per day remained similar, but within the recommended range, with a slight increase from T3 to T12 (T3 17.6 ± 0.85; T12 18.97 ± 0.87, *p* = 0.05). Figure 1.

The additional analyses of HbA1c change over follow up period of one year according to baseline HbA1c values, type of insulin delivery, number of SMBGs per day and duration of diabetes are summarized in Table 2. Seemingly, the change in HbA1c values in the entire cohort was driven by change in the subgroup of subjects with HbA1c ≥ 7% with a drop from 8.22% ± 1.14% to 7.68% ± 1.26% (*p* < 0.0001) in the first 3 months, with no alterations afterwards. On the other hand, in the group of individuals with HbA1c < 7% there was no change in HbA1c, and even a slight increase was noted from 6.26% ± 0.57% to 6.45 ± 0.75%. 

In subjects treated with both CSII and MDI, there was a significant decrease in HbA1c during the first 3 months (7.31% ± 1.17% to 7.1% ± 1.11% vs. 7.51% ± 1.41% to 7.21 ± 1.33%; *p* < 0.05, *p* < 0.001 respectively), while in the follow-up period there was no significant difference in patients treated with CSII. However, in patients treated with MDI, there was a further decline in the period from T3 to T6 (*p* < 0.001), after which the level of HbA1c remained unchanged. CSII subgroup had slightly better glycemic control compared to group treated by MDI (Table 2). 

Significant differences were observed in the regulation of subjects depending on the number of SMGBs per day. The patients who performed SMBG more frequently had lower HbA1c values at baseline compared to those who performed it rarely (7.24% ± 1.89% vs. 7.44% ± 1.06%). In the group of subjects who performed SMBG ≥ 5 times per day, there was no difference in HbA1c during the entire follow-up period, however, in individuals performing SMBG less than 5 times per day, there was a steady decrease in HbA1c levels up to 6 months (*p* < 0.05 and *p* < 0.001, respectively), with stagnation up to T12. Table 2. 

Also, a decrease in HbA1c regardless of the duration of the diabetes in the first three months was observed (*p* < 0.001 for both), while from T3 to T12 no changes were recorded. Table 2. 

As shown in Figure 2, there was a negative correlation between decrease in HbA1c levels and time spent in hypoglycemia (TBR), time spent in range (TIR) and number of scans per day, with positive correlation existent with time spent in hyperglycemia (TAR). This was noted in all time points (T3, T6 and T12) in the entire patient cohort.

## 5. Discussion 

The results of this study confirm what has already been shown in numerous trials [7,8,9,10,11,12,13,14,15]. The simplicity and ease of use of the FGM system which was confirmed by frequent scans on daily basis exactly according to the recommendations, even increasing slightly by the end of one year indicating high level of adherence and compliance resulting in the improvement of HbA1c. Perhaps the most worrisome fact was the average number of SMBGs prior to FGM placement, both in CSII and MDI users, as 5 times per day, pointing out the inconvenience of finger prick methods and poor adherence in performing necessary self-controls with standard glucometers which is in an agreement with the results of previous studies [20,21].

There was a decrease in HbA1c in the entire cohort of patients occurring in the first 3 months with no further changes to the end of one year confirming that majority of improvement in glucoregulation is taking place at the very beginning of FGM implementation in glycemic management, supporting evidence of presently published observational trials [7,8,9,10,11,12,13,14,15]. As the FGM metrics were not available at baseline, data from T3 to 12 were analyzed showing a significant increase in the time spent in range (TIR) and significant decrease in time spent in hyperglycemia (TAR). Apparently, those changes were not profound enough to affect the change of HbA1c in that time period, however correlation analysis revealed negative correlation of HbA1c reduction and TIR and positive correlation with TAR at all time points. So far, only three RCTs reported on TAR and TIR [1,2,3], and only one of them reported a reduction in time spent in hyperglycemia [1], while all of them recorded improvement in percentage of TIR. 

The majority of subjects had symptomatic hypoglycemic events, around half reported nocturnal hypoglycemia and 60% reported asymptomatic hypoglycemic incidents prior the FGM placement. However, from T3 to T12 no significant differences were recorded in time spent in hypoglycemia (TBR), number of hypoglycemic events and duration of hypoglycemia as opposed to previous findings [1,2,3,10,16]. Again, this could be explained by an early effect of the FGM system on glucose management, especially on the reduction of hypoglycaemia occurring in the first couple of months of FSL usage [1]. Conversely, analysis of correlation showed negative correlation of HbA1c decrease and TBR, number and duration of hypoglycemic events present at all time points. There was a decreasing trend observed in TBR, but it is worth noting that percentage of time spent in hypoglycemia was high, around 8%, pointing out difficulties in achieving a percentage below 4% as recommended in the current guidelines [22]. Additionally, our cohort of patients had relatively good glycemic control at the beginning of the study with an average HbA1c of 7.57% ± 1.41%, and at the end 7.16% ± 0.09% which could explain higher TBR since better glucoregulation comes at a cost of higher incidence of hypoglycemia.

The average number of scans per day in our cohort was really high, in average 17 scans per day at T3, remaining the same throughout the whole follow up period indicating that large number of patients followed recommendations given by diabetologist’ team. Also, there was a negative correlation of HbA1c reduction and number of scans per day through all time points. This increase in sensor readings is probably the most important factor for HbA1c improvement as shown in previous studies [18].

The improvement in HbA1c was present in subjects with HbA1c ≥ 7% probably due to higher motivation of poorly controlled individuals to obtain good control of their disease. A well-known fact that the effect of any therapy/intervention on higher HbA1c will be more pronounced than in patients with lower HbA1c values, could also be accountable for this finding. In subjects with HbA1c < 7%, there was no change in HbA1c indicating they were either not sufficiently motivated or the improvement in glucoregulation was limited by the fear of hypoglycaemia. Other RCTs and observational clinical studies also demonstrated that patients with baseline Hba1c > 7.5% FGM usage was associated with a significant reduction in HbA1c, thus further supporting our results [8,10,11,16].

The positive effect on HbA1c levels was recorded regardless of type of insulin delivery, both in CSII and MDI users confirming results of recently published studies [23].

One of the most interesting findings was that subjects who performed SMBG ≥ 5 times per day had no benefits in terms of reducing HbA1c levels, while individuals performing SMBG less than 5 times per day experienced significant decrease of HbA1c in the period of 6 months pointing usefulness of FGM system in obtaining the goal of regular self-control, a crucial factor for successful diabetes management. 

The main limitation of this study is its observational design thus enabling us to analyze changes in parameters of glucoregulation derived from FGM system from T0 to T3, but our intention was to mimic real life clinical settings as much as possible, scheduling follow up visits as usually practiced especially now in the era of the COVID-19 pandemic. 

## 6. Conclusions 

The improvement in HbA1c occurred in the first three months after the FSL placement and was mainly driven by the increase in the number of scans per day. Subjects with poorer glycemic control and those who seldom performed SMBG benefited the most. It is also evident that patients understood and applied given recommendations on glucose management and adjustment of insulin therapy at the beginning of the study depending on the measured glycemia and trend arrows. Therefore, the key point is that the education of patients at the time of placing FGM system provided by the diabetologist’s team was sufficient and successful given that the most profound changes in glucoregulation took place in the first 3 months. Thus, education and motivation seem to be essential, enabling patients to actively participate in decision making based on FGM derived metrics.

## Figures and Tables

**Figure 1 diagnostics-11-00305-f001:**
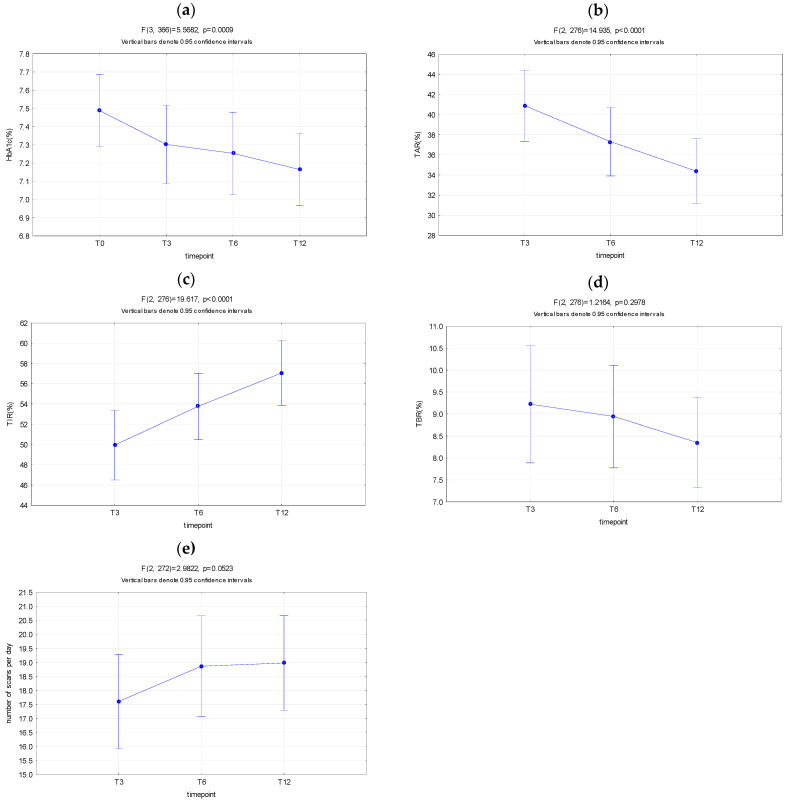
HbA1c, TAR, TIR, TBR, and number of scans over time in the entire cohort. (**a**) HbA1c values significantly decreased from baseline (T0) to 3 months (T3), while no change was observed afterwards. (**b**) Time above range (TAR) decreased significantly over time from T3 to T12. (**c**) Time in range (TIR) increased significantly over time from T3 to T12. (**d**) There was no change in time spent in hypoglycemia (TBR; time below range) over time from T3 to T12. (**e**) Number of scans per day remained similar from T3 to T12, no significant change was observed.

**Figure 2 diagnostics-11-00305-f002:**
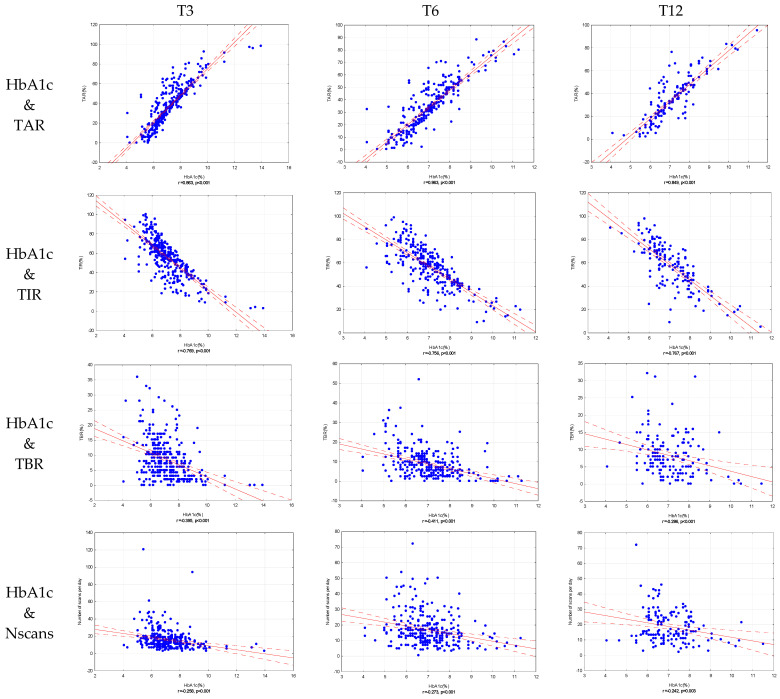
Correlation of HbA1c and parameters of glucoregulation (TAR, TIR, TBR, number of scans per day) at T3, T6 and T12.

**Table 1 diagnostics-11-00305-t001:** Baseline characteristics of the subjects.

**Variable**	**Mean**	**SD**	**Median**	**Minimum**	**Maximum**	***n***
Age years	41.05	14.58	39.0	18.0	81.0	451
Duration of diabetes years	19.88	12.70	18.0	0.5	63.0	437
HbA1c %	7.57	1.41	7.40	4.10	14.2	425
SMBG per day	5.07	2.16	5.0	2.0	20.0	376
**Hypoglycemic Events**	**YES/NO**		***n***		**%**	
Symptomatic hypoglycemic events	YES		410		86.49	
	NO		7		1.47	
Nocturnal hypoglycemic events	YES		248		51.13	
	NO		119		25.1	
Asymptomatic hypoglycemic events	YES		278		58.64	
	NO		126		26.58	

**Table 2 diagnostics-11-00305-t002:** Change in HbA1c over follow up period of one year according to baseline HbA1c values, type of insulin delivery, number of SMBGs per day, and duration of diabetes.

HbA1c%											
Variable	Baseline N 425		3 Months N 320		*p* T3 vs. T0	6 Months N 267		*p* T6 vs. T3	One Year N 147		*p* T12 vs. T6
	M	SD	M	SD		M	SD		M	SD	
HbA1c ≥ 7%	8.22	1.14	7.68	1.26	<0.0001	7.55	1.06	NS	7.55	1.10	NS
HbA1c < 7%	6.26	0.57	6.35	0.73	NS	6.39	0.91	NS	6.45	0.75	NS
CSII	7.31	1.17	7.10	1.11	<0.05	7.03	1.05	NS	7.16	1.02	NS
MDI	7.51	1.41	7.21	1.33	<0.001	7.14	1.19	<0.001	7.14	1.33	NS
SMBG < 5x per day	7.44	1.06	7.25	1.00	<0.05	7.18	1.02	<0.001	7.22	0.93	NS
SMBG ≥ 5x per day	7.24	1.89	6.96	1.65	NS	6.72	1.25	NS	6.80	1.13	NS
Duration of diabetes > 10	7.44	1.27	7.25	1.29	<0.001	7.12	1.14	NS	7.13	1.07	NS
Duration of diabetes < 10	7.65	1.60	6.88	1.12	<0.001	7.12	1.21	NS	7.12	1.16	NS

## Data Availability

Data is contained within the article.

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
