# Peer review of "Assessment of FreeStyle Libre Flash Glucose Monitoring System Implementation in Real Life Clinical Setting: A Prospective Observational Study"

_diagnostics, 2021, doi:10.3390/diagnostics11020305_

Round 1

Reviewer 1 Report

This observational studies on the effectiveness of FreeStyle Libre flash glucose monitoring system implementation in real life clinical setting in 451 subjects with Type 1 diabetes over 3-12 months validates the hypothesis that improvement in HbA1c was mainly driven by the increase in the number of scans per day. The flash glucose monitoring (FGM) benefits subjects with poorer glycemic control and  those who seldom performed SMBG benefited the most.

This study emphasises on the education of subjects with T1D at the time of placing  FGM system provided by the diabetologist' team.

My only suggestion is to replace "patients" with "subjects of Type 1 diabetes".

Author Response

My only suggestion is to replace "patients" with "subjects of Type 1 diabetes."

Changes have been made according to the suggestion. 

Reviewer 2 Report

The paper is well written and can be interesting to support the use of FGM systems but no new findings are reported.

Please modify the paper according to the following comments:

  1. The title is very ambitious, please to tone down the title.
  2. In the abstract and introduction, the authors emphasize in the education stage, but no quantitative or qualitative information supports such assertions. Please remove that or provide accurate information to conclude that education was crucial.
  3. In the abstract, authors claim that 451 patients were evaluated in a period of 12 months, but actually only 425 were recruited and only 147 were followed up to 12 months. This information is misleading, please be more accurate and coherent throughout the paper.
  4. The information about clinical protocols and Institutional Review Board is not complete. It is very desirable that information can be free accessed by the readers.

Author Response

The title is very ambitious, please to tone down the title.

The title has been changed to Assessment of FreeStyle Libre flash glucose monitoring system implementation in real life clinical setting: a prospective observational study.

In the abstract and introduction, the authors emphasize in the education stage, but no quantitative or qualitative information supports such assertions. Please remove that or provide accurate information to conclude that education was crucial.

The following sentence “Education of patients at the time of placing FGM system provided by the diabetologist' team was crucial” has been deleted from the Abstract section.

In the abstract, authors claim that 451 patients were evaluated in a period of 12 months, but actually only 425 were recruited and only 147 were followed up to 12 months. This information is misleading, please be more accurate and coherent throughout the paper.

This has been corrected in the Abstract section and  Method section .

The information about clinical protocols and Institutional Review Board is not complete. It is very desirable that information can be free accessed by the readers.

Number of Ethics Committee approval has been inserted in the Method section - No: R2-12487/2019.